# Optimal Resource Allocation for 5G Network Slice Requests Based on Combined PROMETHEE-II and SLE Strategy

**DOI:** 10.3390/s23031556

**Published:** 2023-01-31

**Authors:** Sujitha Venkatapathy, Thiruvenkadam Srinivasan, Han-Gue Jo, In-Ho Ra

**Affiliations:** 1School of Computer Information and Communication Engineering, Kunsan National University, Gunsan 54150, Republic of Korea; 2School of Electrical Engineering, Vellore Institute of Technology, Vellore 632014, Tamilnadu, India

**Keywords:** 5G network, virtual network embedding, resource allocation, heuristic fuzzy, shortest path

## Abstract

The network slicing of physical infrastructure is required for fifth-generation mobile networks to make significant changes in how service providers deliver and defend services in the face of evolving end-user performance requirements. To perform this, a fast and secure slicing technique is employed for node allocation and connection establishment, which necessitates the usage of a large number of domain applications across the network. PROMETHEE-II and SLE algorithms were used in this study’s approach to network design for node allocation and link construction, respectively. The PROMETHEE-II approach takes into account a variety of node characteristics while constructing a node importance rank array (NIRA), including the node capacity, bandwidth of neighboring connections, degree of the node, and proximity centrality among others. The SLE method is proposed to record all possible link configurations for the network slice request (NSR) nodes to guarantee that the shortest path array (SPA) of the NSR has a high acceptance rate. Performance metrics such as the service revenue and acceptance ratio were considered to evaluate the effectiveness of the suggested approach. The effectiveness of network slicing has been further examined under different infrastructure models to determine whether a small-world network structure is beneficial to 5G network. For each scenario, simulations were carried out and the results were compared to previously published findings from other sources.

## 1. Introduction

In recent years, private and commercial 5G mobile networks have faced enormous mobile traffic due to a growth in the use of portable devices by consumers. In the near future, 5G technology is projected to assist many new enterprises and vertical sectors, including transportation, healthcare, and the energy supply chain. A highly sophisticated 5G architecture [1] is required for providing high-level, continuous, and reliable service to user equipment (UE) with a broad variety of needs. The next generation mobile networks (NGMN) by Alliance established the network slicing idea to achieve resource allocation for user devices with varying performance needs in 5G network settings [2]. Individual slices of physical infrastructure are created based on the grouping of requirements through a set of virtual network functions (VNFs) that have virtual resources, logical topology, traffic regulation, and node and link provisioning rules, as well as security monitoring parameters to guarantee the quality of service (QoS) [3]. The 5G system architecture that supports network slicing was previously established in the first edition of 5G normative standards and was authorized by the 3rd Generation Partnership Project (3GPP) [4]. According to the physical architecture, each logical network slice is generally made up of three key components: the radio access network (RAN) and the core network (CN) [5]. The logical networks are shown in Figure 1 with the components separated, assuming that the physical infrastructure is being used as a shared resource. RAN behavior was investigated in the work [6] by taking into account the criteria associated with the quality of service in order to provide a dependable service to the UE. Nojima et al. [7] dealt with network isolation in their study, and they allocated resources using traditional packet scheduling and dynamic resource block allocation techniques. In an orthogonal frequency division multiple access (OFDMA) context, network slicing and resource allocation based on a sub-carrier and power were performed in [8], where the maximization of the network summation rate was treated as the objective function, with the traffic rate of each slice assumed. In [9], network slices were found to be chains of virtual network functions and real resources to meet the varying needs of end users, leading a variety of studies on the issue of VNF deployment and orchestration.

In [10], the VNF orchestration problem was framed as an integer linear program and addressed using a heuristic method, with the cost function consisting of the VNF deployment cost, the energy cost, the traffic cost, and the propagation delay cost. Another solution, known as WiNE (wireless network embedding) [11], was developed to address the issue of VNF placement in large-scale installations. For the purpose of solving the VNF placement issue according to requests, WiNE examines three various situations, including linear VNF requests, branching VNF requests, and VNF requests containing loops, among others. Using software-defined networking (SDN) and VNF together was detailed in [12] to provide network slicing, which also covered the underlying principles, designs, and challenges of the two techniques. Recent research shows that VNF-based network slicing is superior to SDN when it comes to increasing the profit and maximizing the utilization of resources. A thorough examination of the strategies for providing logical network nodes and connections from physical resources, which is referred to as the virtual network embedding (VNE) issue, is provided in [13]. VNE is also known as network slice provisioning, and [14] provides a framework for allocating physical resources to logical networks. According to previous network slicing research findings, network slice provisioning becomes difficult in many ways, including ensuring the highest possible acceptance ratio, secure data transmission, and a low latency, and providing service for a diverse range of user requirements [15,16].

In response to the concerns identified in the literature, a hybrid approach to node slice provisioning was devised. The physical network architecture was also designed to make the network slice more user-friendly. The PROMETHEE and SLE algorithms were used in this research to allocate nodes and create connections for the logical networks inside the physical network. This project was completed in two independent stages. First, a node importance ranking array (NIRA) was constructed using PROMETHEE-II and various node attributes, and then nodes were assigned according to the NIRA. This was performed in the following step, while the SLE algorithm establishes a link between the targeted nodes. The following are the primary contributions of the planned work:A mathematical model is introduced for network slicing that takes operational constraints and security considerations into account when assigning nodes and connecting them.The prepared node importance rank array (NIRA) was used to allocate NSR nodes in physical and logical networks, and the SLE approach was utilized to connect the network slice request (NSR) nodes. NIRA preparation takes into account node information such as the node capacity, nearby line bandwidth, degree of the node, and the node’s proximity centrality.A PROMETHEE-II multi-criteria approach is suggested for NIRA preparation, and an SLE algorithm is provided for NSR node link creation. The SLE approach generates the shortest path array (SPA) for NSR nodes, guaranteeing that all possible connections are installed and increasing the acceptance percentage.The proposed PROMETHEE-SLE approach was evaluated on a small-world network to confirm that the infrastructure is slicing-friendly, and the results were also compared to those obtained from a scale-free physical infrastructure.

## 2. Related Works

The current efforts for optimum resource allocation in 5G network slicing, as well as their extensive performance comparisons and strategies used, are discussed in this section. VNE methods from the ViNEYard (virtual networks embedding yard) library, D-ViNE, and R-ViNE (deterministic and random rounding) were used in [17] to solve a network resource allocation problem adopted as a mixed integer program via substrate network augmentation. The concepts of virtualization and service orientation that are employed in networking technology make cloud-based networking viable [18]. The authors of [19] presented a deployment method based on complex network theory, with the goal of optimizing the resource efficiency and acceptance ratio while minimizing costs. It is recommended that two phases are completed in the proposed method: (i) the installation of virtual network functions and (ii) the selection of link route options. A further evaluation of the suggested approach’s efficacy was carried out by comparing the solution to those obtained with simulated annealing (SA) [20], the VNE algorithm based on LAVA [21], and VNF placement achieved by the GLL method [22]. Ref. [23] deployed VNFs using network topology and resource-based algorithms (VNE-NTANRC).

All substrate and virtual nodes were ranked using computed node values (NoV), which are defined by network topology characteristics and global network resources. In addition to the average VNR acceptance ratio, the method’s performance was measured by average node and link utilization. For the purpose of improving the quality of service, a slice-based physical resource sharing scheme was implemented in a non-orthogonal multiple access system (NOMA) [24]. The optimization of the total user rate remains the objective function in this technique, which is expressed as a Markov decision process issue for the allocation of subcarriers. In [25], network slicing for enhanced mobile broadband (eMBB) and ultra-reliable low-latency communication (URLLC) requests was provided by considering optimizing the spectral efficiency as the goal function that is subject to the reliability constraint. Using the orthogonal frequency division multiple access (OFDMA) technology, the slice model was created, and the problem was addressed using the joint power and subcarrier allocation (JPSA) approach. Adaptive particle swarm optimization was used to demonstrate the usefulness of the suggested technique (APSO) [26]. For the production of node ranking, the VIKOR technique, which is one of the multi-criteria decision-making approaches, was introduced in [27]. The provisioning of virtual network nodes was carried out in accordance with the node ranking that had been developed for the physical and logical networks. It was decided to use Floyd’s method for link provisioning in order to offer the shortest route for the NSR nodes.

Recently, machine learning and deep learning methods have been applied to the problem of resource allocation in virtual networks [28,29]. Slicing categorization, data collection, and slicing extraction were all part of the process used to create network slices. GS-DHOA (glow-worm swarm-based deer hunting optimization algorithm) was also used to improve the weight function of the training network [28]. The provisioning of nodes and connections for virtual networks in physical networks was carried out in accordance with the categorization results of the virtual networks. A scalable digital twin (DT) for network slicing is presented in [29], with the goal of capturing and monitoring the end-to-end (E2E) metrics of slices across a wide range of network settings while maintaining scalability. Table 1 summarizes the literature’s discussion of the objectives, node and link provisioning procedures, security problems, and physical network.

The acceptance rate and resource efficiency are the fundamental objectives of the majority of research investigations. The distinction between the implemented approaches is based only on the process for deploying VNFs. In this suggested research, PROMETHEE-II and SLE algorithms were employed to install VNFs in the physical infrastructure. Additionally, almost all of the methodologies for network slicing make use of a scale-free network model [30], which has some drawbacks in terms of identifying the shortest pathways for NSRs. Scale-free network models will be less efficient when NSRs are more closely scheduled and also have a longer life-time. This initiative aims to construct a slice-friendly physical infrastructure in order to maximize benefits to stakeholders with the fewest possible facilities. The small-world network is one of the physical network infrastructures proposed for future wireless mobile networks. Experiments in [31] were performed to make a complete investigation of the small-world network. This research adopted the Watts–Strogatz [32] small-world network concept to develop a slicing-friendly physical design. The most important breakthrough in this work is the utilization of the small-world network for the construction of physical infrastructure. Utilizing a small-world network results in improvements to the bandwidth, as well as CPU resources.

## 3. Proposed System Model

### 3.1. Physical Infrastructure Model

The most important aspect of network slicing is the physical infrastructure. The Watts–Strogatz [32] small-world network model is best suited for a network with a fixed number of nodes. Small-world networks feature sub-networks with connections between any two nodes that have a high clustering coefficient, allowing for many pathways between nodes. This network model is built in two phases, which are as follows:

**Step 1**: Create an undirected graph with *N* nodes linked to *M* nearby nodes, assuming M/2 on either side of nodes. Any edge (*i*, *j*) might be created in a network of nodes (0,1,2,3……N−1) only if it satisfies the following condition:(1)$$0<|i-j|mod\left(N-1-\frac{K}{2}\right)\leq \frac{K}{2}$$

**Step 2**: Rewire the probability β for each edge (i,jmodN) with $i<j\leq i+\frac{K}{2}$. This is accomplished by substituting (i,jmodN) for (i,k), where *k* is evenly picked at random from all nodes. The value of β is considered as any number between 0 and 1. The foregoing two procedures result in the creation of an undirected physical network with *N* nodes and NK/2 connections, $G_{P}=\left(N_{P},E_{P}\right)$, where $N_{P}$ indicates the set of nodes and $E_{P}$ denotes the set of linkages in the physical structure. As with earlier research, this approach evaluates the node capacity and security power level for each node. The capacity of a node is determined by the number of CPUs that are available. The total number of CPUs available in the $i^{th}$ node is defined as $CPU_{available,i}$. The security of the $i^{th}$ physical node is denoted by $SPL_{i}$, which represents the highest level of security that a physical node may provide to the NSR node. The primary parameter for connections linking nodes *i* and *j* is the link’s bandwidth, $BW_{available,ij}$. The lengths of the various connections are supplied in order to discover the shortest route between any two nodes. For example, the variable $L_{i,j}$ stores the length of the line between the nodes *i* and *j*.

### 3.2. NSR Model

This study made the assumption that each NSR requires the specified number of nodes, CPU capacity, bandwidth, security, and lifetime. Hence, the NSR request was modeled as an undirected graph $G_{NSR}=\left(N_{NSR},E_{NSR}\right)$  where $N_{NSR}$ indicates the set of nodes in NSR and $E_{NSR}$ denotes the set of linkages in NSR. Each node consists of $\left(CPU_{NSR},SPL_{NSR},LT_{NSR}\right)$, where $CPU_{NSR}$ is the CPU capacity of the NSR. $SPL_{NSR}$ is the security power level of the associated NSR. $LT_{NSR}$ is the lifetime of the NSR. $BW_{NSR}$ is the set of bandwidth requirements of requested links in NSR [33]. During each interval, the physical infrastructure was used to allocate nodes and establish links for the NSR. Each time a new NSR is received, the physical infrastructure’s available resources were updated depending on the preceding request’s $LT_{NSR}$.

### 3.3. NSR Deployment Strategy

Each NSR was deployed on the physical network in two successive steps. First, the NSR’s nodes were allocated in accordance with the NIRA developed for both physical and logical networks. Second, the creation of links was carried out based on the SPA through SLE. Basically, the NIRA is a dynamic array that must be updated once each slice request is processed. Link mapping begins once nodes are allocated through the mapping of NIRAs for the physical and logical networks. SPA is created for link mapping using the SLE method, which employs Dijkstra’s algorithm [34] to determine the shortest route between two nodes. Figure 2 illustrates the phases involved in the NSR deployment method.

NSR Acceptance Ratio (NAR): This metric provides the direct measurement of the adapted network slicing technique on the given physical infrastructure. It is measured by taking the ratio between successfully completed NSRs and unsuccessful NSRs for a given time $T_{max}$, which is expressed as
(2)$$P_{NAR}=\frac{\sum_{t=0}^{T_{max}}NSR_{success}\left(t\right)}{\sum_{t=0}^{T_{max}}NSR_{unsuccess}\left(t\right)}$$
where $NSR_{success}\left(t\right)$ and $NSR_{unsuccess}\left(t\right)$ are served and unserved NSR at time *t*, respectively.Resource Efficiency (RE): This metric is measured by calculating the achieved revenue and the investment cost made for providing the physical infrastructure. The revenue can be determined from the CPU capacity of nodes and link bandwidth requested for the NSRs. The investment cost is estimated from the physical infrastructure for the case. The expression for calculating RE is given by
(3)$$P_{RE}=\sum_{t=0}^{T_{max}}\frac{NSR_{revenue}\left(t\right)}{PSI_{cost}\left(t\right)}$$
(4)$$NSR_{revenue}\left(t\right)=CPU_{requested,t}+BW_{requested,t}$$
(5)$$PSI_{cost}\left(t\right)=CPU_{requested,t}+BW_{requested,t}.L_{t}$$
where $NSR_{revenue}\left(t\right)$ is the revenue of NSR at time *t*, $PSI_{cost}\left(t\right)$ is the utilized physical infrastructure for the NSR at time *t*, $CPU_{requested,t}$ and $BW_{requested,t}$ are the requested numbers of CPU and BW of NSR at time *t*, and $L_{t}$ is the length of the shortest path for NSR at time *t*.Problem Formulation: It is understandable that effective network slicing results in the maximum utilization of physical network resources while minimizing slice provisioning costs [27]. As a result, the problem of minimizing the cost of slice provisioning is formulated using the integer linear programming model in conjunction with the necessary constraints as shown below.
(6)$$min,\sum_{N_{k,NSR}\in N_{NSR}}\sum_{N_{k,P}\in N_{P}}b_{i,k}(1+SPL_{i,P}\left(CPU_{i,NSR}\right)+\sum_{E_{kl,NSR}\in E_{NSR}}\sum_{E_{ij,P}\in E_{P}}a_{ij,kl}BW_{kl,NSR}$$
(7)$$Subjectto,\sum_{N_{i,P}}x_{i,k}=1,\forall N_{i,P}\in N_{P}$$
(8)$$\sum_{N_{j,NSR}}x_{i,k}\leq 1,\forall N_{j,NSR}\in N_{NSR}$$
(9)$$x_{i,k}CPU_{k,NSR}\leq CPU_{i,P},\forall N_{i,P}\in N_{P},\forall N_{j,NSR}\in N_{NSR}$$
(10)$$x_{i,k}SPL_{k,NSR}\leq SPL_{i,P},\forall N_{i,P}\in N_{P},\forall N_{j,NSR}\in N_{NSR}$$
(11)$$\sum_{N_{ij,P}}\left(a_{ij,kl}-a_{ji,kl}\right)=x_{i,k}-x_{i,l},\forall N_{i,P}\in N_{P},\forall E_{kl,NSR}\in E_{NSR}$$
(12)$$\sum_{N_{ij,P}}a_{ij,kl}BW\left(E_{kl,NSR}\right)\leq BW\left(E_{ij,P}\right),\forall E_{ij,P}\in E_{P}$$$b_{i}^{k}\in \left(0,1\right)$ is a binary variable, where $b_{i}^{k=1}$ indicates that $N_{NSR}$ is served onto $N_{P}$, and is otherwise 0. $a_{ij,kl}$ indicates whether the link $E_{ij,P}$ hosts the request link $E_{kl,NSR}$: the value will be 1 if the link exists; otherwise, it will be 0. Network providers use some additional resources to ensure the security level of nodes. Along with CPU capacity and bandwidth, we also considered the link capacity, delay rate, and jitter resources to improve the allocation of resources and accuracy of node slice creation. Constraint (7) guarantees that each request node is allocated onto the physical node. Constraint (8) ensures that the physical node can only host one node from the same request. The CPU capacity is represented in constraint (9). Constraint (10) ensures the security constraints of each node. Constraint (11) indicates the flow passing through physical nodes. Constraint (12) guarantees that the bandwidth requested by the NSR link should not exceed the available bandwidth in the physical link. The objective of this work was to improve the use of resources and the adoption rate of network slicing. Physical infrastructure restrictions were kept in mind as part of this solution method. Every NSR requirement is guaranteed to be within the stated limits given below:
(13)$$N_{min}\leq N_{NSR}\leq N_{max}$$
(14)$$BW_{min}\leq BW_{NSR}\leq BW_{max}$$
(15)$$CPU_{min}\leq CPU_{NSR}\leq CPU_{max}$$
(16)$$LT_{min}\leq LT_{NSR}\leq LT_{max}$$
(17)$$SPL_{min}\leq SPL_{NSR}\leq SPL_{max}$$
(18)$$T_{max}>LT_{max}$$
where $N_{min}$ and $N_{max}$ are the minimum and maximum number of nodes per NSR, respectively, $BW_{min}$ and $BW_{max}$ are the minimum and maximum number of BW per NSR, respectively, $CPU_{min}$ and $CPU_{max}$ are the minimum and maximum number of CPUs per node per NSR, respectively, $LT_{min}$ and $LT_{max}$ are the minimum and maximum life time per NSR, respectively, and $SPL_{min}$ and $SPL_{max}$ are the minimum and maximum security level per node per NSR, respectively. The limitations specified in Equations (12)–(16) correspond to the NSR limits on the node count, bandwidth, CPU capacity, security level, and life duration. Constraint (17) ensures that the NSR’s lifetime $LT_{NSR}$ does not exceed the overall transmission time interval $T_{max}$.

### 3.4. Factors for Node Importance Calculation

This problem takes into account four crucial aspects that might have an impact on every node in a complex network. The NIRA of nodes in a network is created depending on the value of the variables that have been determined for them. It is derived from the basic complex network theory that the formulations for the factors are derived.

Node Capacity Factor (NCF): The availability of the number of CPUs on the node is used to determine the capacity of the node. A node with a higher number of CPUs may give service to a greater number of NSRs. As a result, a node with a large number of CPUs should be given a better ranking in the NIRA system. The value of the factor is updated immediately upon the accomplishment of each NSR as below:
(19)$$f_{NCF,i}\left(t\right)=CPU_{available,i}\left(t\right)-CPU_{NSR,i}\left(t\right)$$
where $CPU_{available,i}\left(t\right)$ is the available CPU capacity of the $i^{th}$ node of physical infrastructure at time *t*, and $CPU_{NSR,i}\left(t\right)$ is the requested CPU capacity of the $i^{th}$ node of the NSR received at time *t*.Node Topology Factor (NTF) and Node Bandwidth Factor (NBF): In both cases, the number of adjacent connections present with the physical node determines the NTF and NBF. The total number of adjacent connections of a node is represented by NTF, whereas the total number of adjacent links bandwidth is represented by NBF. Any node with a greater NTF and NBF is given a higher priority in the NIRA. Each NSR results in an update of the factors. The equations for updating the factors are shown below:
(20)$$f_{NTF,i}\left(t\right)=\sum_{j=1}^{tNodes}a_{ij,available}\left(t\right)$$
(21)$$f_{NBF,i}\left(t\right)=\sum_{j=1}^{tNodes}\left(BW_{available,ij}\left(t\right)-BW_{NSR,ij}\left(t\right)\right)$$
where tNodes is the total nodes present in the network, $a_{ij,available}\left(t\right)$ is 1 if the $i^{th}$ node has a connection with $j^{th}$ node and unserved at time *t*, and 0 otherwise, $BW_{available,ij}\left(t\right)$ is the available bandwidth of the link connecting node *i* and node *j* of physical infrastructure at time *t*, and $BW_{NSR,ij}$ is the requested bandwidth of the link connecting node *i* and node *j* of the NSR at time *t*.Node Closeness Centrality Factor (NCCF): Using the factors NCF, NTF, and NBF, we may learn about the node’s local information. A node’s strength in a network should be measured by how much it can influence the shortest path. A node’s global relevance is determined by its shortest route information, which is provided by NCCF. Therefore, the closer a node is to other nodes, the greater its centrality. Listed below is the NCCF equation at time *t*.
(22)$$f_{NCCF,i}\left(t\right)=\{\sum_{j=1}^{nNodes}L_{i,j}{\}}^{-1},i\neq j$$
where $L_{i,j}$ is the shortest path length between node *i* and *j*.

## 4. Proposed Strategies for NSR Deployment

### 4.1. PROMETHEE-II Strategy for NIRA Preparation

The preparation of the NIRA was used to carry out the node allocation of the logical structure on the physical structure in this piece of work. The NIRA for physical infrastructure was developed before receiving the NSR, and the same for the logical structure was prepared upon receipt of the NSR, respectively. The value of each node was determined by a variety of elements engaged in the network connecting to the node, including the NCF, NTF, NBF, and NCCF. Consequently, it is critical to construct the ranking without compromising the specifics of the node’s configuration. A famous multiple-criteria decision-making approach established by Brans [35] was utilized for NIRA preparation, and PROMETHEE-II was one of the methods employed. More precisely, the preferences of and differences in the elements were taken into account by this technique. The following are the stages that are involved in the PROMETHEE-II approach:

**Step 1:** Preparation of evaluation table. The evaluation table keeps the values of factors for each node. Therefore, the factors of each node are evaluated with their respective weight parameters for preparing Table 2. It is assumed that $N=n_{1},n_{2},\ldots n_{i}$ is a set of nodes and $F=f_{1},f_{2},f_{3},f_{4}$ is a set of factors.

**Step 2:** Determination of preference function. This value is determined by the pairwise comparison between all of the nodes for each factor of a network.
(23)$$d_{k}\left(n_{i},n_{j}\right)=f_{k}\left(n_{i}\right)-f_{k}\left(n_{j}\right)$$
where $d_{k}\left(n_{i},n_{j}\right)$ refers to the difference between the evaluations of two actions for factor $f_{k}$. The preference function is introduced to convert the $d_{k}\left(n_{i},n_{j}\right)$ unified value to the following:(24)$$P_{k}\left(a,b\right)=G_{k}\left[d_{k}\left(n_{i},n_{j}\right)\right]$$

**Step 3**: Determination of global preference index.
(25)$$\pi \left(n,m\right)=\sum_{k=1}^{i}P_{k}\left(a,b\right).w_{k}$$
where $w_{k}$ is a weight function of factor *k*, which is assumed to be greater than 0, and the sum of the weights is equal to 1.

**Step 4:** Determination of positive and negative outranking flows. The following two factors are calculated to locate each node with respect to all other nodes.
(26)$${Ø}^{+}\left(a\right)=\frac{1}{\left(i-1\right)}\sum_{x\in A}\pi \left(a,x\right)$$
(27)$${Ø}^{-}\left(a\right)=\frac{1}{\left(i-1\right)}\sum_{x\in A}\pi \left(x,a\right)$$
where ${Ø}^{+}\left(a\right)$ is obtained by ranking the nodes according to the non-increasing values of their respective positive flow values. and ${Ø}^{-}\left(a\right)$ is obtained by ranking the nodes according to the non-decreasing values of their respective negative flow values.

**Step 5:** Determination of net flow.
(28)$$Ø\left(a\right)={Ø}^{+}\left(a\right)-{Ø}^{-}\left(a\right)$$

The highest value of Ø(a) refers to the best node in NIRA. Algorithm 1 is used to explain the process of node allocation through NIRA.
**Algorithm 1:** Node allocation through PROMETHEE-II**Input:**GP,GNSR**Output:**Nodeallocation**for** each node NP(i) of GP **do**   Calculate fNCF,i,fNTF,i,fNCCF,i,fNBF,i   Ø(i) for Physical infrastructure using PROMETHEE-II**end for****for** each node NNSR(j) of GNSR **do**   Calculate fNCF,j,fNTF,j,fNCCF,j,fNBF,j   Prepare Ø(j) for NSR using PROMETHEE-II**end for**Set allocationTrack=0**for** each node Ø(i) of GP **do**   **if** Ø(j) is not empty **then**     **for** each node Ø(j) of GNSR **do**        **if** CPUNSR,j<CPUavailable,i and BWrequested(k)<BWavailable(l) **then**          node Allocation NP(l) to NNSR(k)          Update CPUserved,l and alj,served∀NP(l) of GP and BWlj,served∀EP(l,j)          Update fNCF,l,fNTF,l,fNCCF,l,fNBF,l        **else**          Move to next node        **end if**     **end for**   **else if** allocationTrack=NNSR
**then**     Return NSR node allocation unsuccessful   **else**     Return NSR node allocation success   **end if****end for****return**
nodeAllocation


### 4.2. SLE Strategy for SPA Preparation

Identifying the shortest route for the NSR nodes in the physical infrastructure is the primary goal of this SLE technique. Regarding NSR, it is clear that sometimes the most direct route is not the best one. In certain cases, a connection in the chosen shortest route may be connected to the already supplied NSRs. This makes an identification of all of the connections linking NSR nodes and organizing them in SPA in a non-increasing order according to the length of their individual pathways very necessary. This array may be used for the establishment of a connection for NSRs. It is anticipated that the shortest pathways indicated would be referred to one after another throughout the connection building process. The SPA chooses the path with the shortest length among those that are not engaged for link provisioning. Algorithm 2 is used to explain the process of preparing SPA.
**Algorithm 2:** SLE for SPA preparation1:**Input:**$G_{P}=\left(N_{P},E_{P}\right),L_{i,j}$ of all $E_{P}$2:**Output:**SPA3:Set, SN: source nodeDN: destination node*K*: the number of shortest paths between SN and DN$P_{u}$: a path from SN to *u*SPA: a heap structure keeps paths*P*: set of shortest paths from SN and DNcountu: number of shortest paths for node *u*4:Initialize, P=empty,countu=0,L=05:**for** all *u* node in $N_{P}$ **do**6:   Insert path $P_{SN}=SN$ into SPA with cost 07:**end for**8:**while**SPAxxxempty&&countDN<K**do**9:   SPA=SPA−Pu10:   countu++11:   **if** u=DN **then**12:     P=PUPu13:   **end if**14:   **if** countuwwwK **then**15:     **for** each $E_{P}$ adjacent to node *u* **do**16:        $P_{v}$ new path with $L=L+L_{i},u$17:        Insert $P_{v}$ into SPA18:     **end for**19:   **end if**20:**end while**21:**return**SPA

### 4.3. Hybrid PROMETHEE-II and SLE Approach for Deployment

This operation allots a node and establishes a connection between the nodes of the NSRs that have been received during the time interval from 0 to $T_{max}$. Throughout this study, it was assumed that all of the required NSR nodes belong to a single category, such as eMBB, mMTC, or URLLC. A NSR should be able to request any slice type at any point in time. The PROMETHEE-II method was used to allocate nodes for the NSRs, and the SLE algorithm, which is based on Dijkstra’s shortest route algorithm, was used to construct links between them. The hybrid PROMETHEE-II and SLE algorithms were used to obtain the best possible optimum provisioning for the NSR. In the physical infrastructure, the NIRA and SPA were updated in accordance with the lifetime of the received NSR, which allows for the identification of the most appropriate and optimum provisioning for the next arrival of NSR. As a result, the dynamic allocation of NSRs in the physical infrastructure was made effective and efficient. The success rate and resource efficiency of slice provisioning are two major indicators that are used to evaluate the efficacy of slice provisioning. This is accomplished by Algorithm 3 for the proposed hybrid algorithm, which allocates resources for NSRs for the maximum amount of time $T_{max}$. The flowchart in Figure 3 depicts the sequence of events that occur throughout the whole solution method.
**Algorithm 3:** NSR deployment through hybrid PROMETHEE-II and SLE1:**Input:** NSR deployment2:**Output:**SPA3:Set, $CPU_{served,i}=0$ and $a_{ij,served}=0,\forall N_{P}\left(i\right)$ of $G_{P}$ and$BW_{ij,served}=0,\forall E_{P}\left(i,j\right)$ at t=04:**while**$t<T_{max}$**do**5:   Get $G_{NSR}$6:   Call Algorithm 1 for Node Allocation7:   **if** NSR node allocation unsuccessful **then**8:     Increment $NSR_{unsuccessful}\left(t\right)$9:   **else**10:     Increment $NSR_{success}\left(t\right)$11:     Call Algorithm 2 for SLE12:     Calculate $NSR_{revenue}\left(t\right)$ and $PSI_{cost}\left(t\right)$13:   **end if**14:   Increment *t*15:**end while**16:Calculate $P_{NAR}$ and $P_{RE}$

## 5. Simulation Results and Discussions

The suggested technique was validated using two distinct scenarios: (i) an implementation of the proposed hybrid PROMETHEE-II and SLE algorithm for NS, and (ii) an implementation of the algorithm with varying physical infrastructure provisions. The resource efficiency and acceptance ratio of the scenarios were used to determine their level of competence. The factors taken into consideration for the test case are included in Table 3, which covers the range of available resources in the physical infrastructure, as well as the range of NSRs needs for the test scenario. In order to build the simulation platform, we used the Python programming language.

### 5.1. Implementation of Proposed Algorithm for NSR Resource Allocation

According to the Watts–Strogatz model, physical infrastructure is developed in order to execute the suggested algorithm. This infrastructure is based on the small-world network. The competence of the suggested technique was evaluated in comparison to works published in the literature, including NSR-NR [19], SA [20], LAVA [21], GLL [22], and VIKOR [27]. In order to ensure that the results of different algorithms are more comparable, the same parameters were used in all of them. The NS’s resource efficiency and acceptance ratio were taken into account while evaluating the algorithms’ performance. This study evaluated the number of nodes required by NSRs in various categories, such as 10, 20, and 30 in the context of the physical infrastructure availability of 100, 200, and 300 nodes [19,22,27]. All network features, such as the CPU capacity of each node, security level of each node, bandwidth of the connections, and length of the lines, had their respective values chosen at random from a pool of values within the defined range.

The resource efficiency and acceptance ratios achieved for the various algorithms under various operating situations are shown in Figure 4 and Figure 5. To facilitate a comparison, several techniques were used to allocate nodes and connections for various NSRs. Regarding the overall observation regarding resource efficiency and NSRs, it is evident from Figure 4 that the network’s resource efficiency declines as the number of NSRs rises, whereas the number of nodes in the physical infrastructure enhances the efficiency. As can be seen from the resource efficiency equation, increasing the number of nodes in a network reduces the shortest route length, resulting in a better use of available resources. Additionally, the increased number of NSRs necessitates the allocation of additional nodes with the needed CPU capacity and bandwidth. The proposed method achieves a resource efficiency of 0.73, 0.68, and 0.66 for NSRs 10, 20, and 30, respectively, for an infrastructure equipped with 100 nodes, as shown in Figure 4a. As seen in Figure 4b,c, the efficiency rises as the number of nodes grows to 200 and 300, resulting in a maximum efficiency of 0.9 under 10 NSRs for a 300-node physical resource. Additionally, it supports a maximum of 17,554 CPUs and a minimum of 2921 CPUs within the infrastructure of 100 and 300 nodes, respectively. Additionally, a maximum of 16,676 and a minimum of 2775 bandwidth are used under the same operating state. When comparing the performance of the methods, the suggested approach outperforms the others under a variety of operating situations and with variable NSRs.

Among the various methods, the technique based on VIKOR achieves the best results since it employs MCDM-based node ranking for node provisioning and Floyd’s shortest route algorithm for link provisioning. In comparison to all other algorithms, the technique based on GLL has the lowest performance. The other algorithms, such as NSR implementation, an SA-based method, and LAVA, all outperformed GLL. When a network is equipped with 100 nodes and requires 30 NSRs, the GLL algorithm records 0.44 as the minimum value for resource efficiency. The suggested technique for the acceptance ratio was analyzed using various values of NSRs ($T_{NSR}$) ranging from 5 to 35. The NSRs’ lifetimes were chosen at random from the defined range. The findings obtained after the algorithms were successfully executed under various physical infrastructure conditions and are shown in Figure 5.

The data demonstrate that, as the ($T_{NSR}$) value is raised, the acceptance ratio decreases even though the number of physical nodes is increased. However, having more nodes in the physical structure results in a much higher acceptance percentage. Additionally, the acceptance ratio increases when the life of the NSRs decreases. The proposed technique achieves a maximum acceptance ratio of 0.99 when ten NSRs are used in conjunction with the 300 accessible nodes. The minimal acceptance ratio is 0.687 when 35 NSRs are used in conjunction with 100 accessible nodes. In contrast to previous algorithms, the suggested approach outperforms them under a variety of operating situations. Among the five algorithms, the technique based on VIKOR achieves the second-best result, which is comparable to the PROMETHEE-II-SLE algorithm’s findings. The lowest acceptance ratios are 0.5848 and 0.6069 for 35 NSRs under 100 available nodes, respectively, when network slicing is performed using the LAVA and GLL approaches.

### 5.2. Implementation of the Algorithm under Different Physical Infrastructure Networks

The effectiveness of the proposed algorithm was further validated with different working conditions under two different physical infrastructure networks. We considered a scale-free network (SFN) and small-world network (SWN) for the performance investigation. The characteristic behaviors of the two networks are entirely different.

The construction of the SWN and SFN was carried out in this study by using the Watts–Strogatz model and the Barabsi–Albert model [30], respectively. The comparison of the two networks has been carried out for many decades, and the findings show that the SWN has more benefits than the SFN since it can connect practically any two nodes in a network, but the SFN cannot. The purpose of this research was to determine the efficacy of the SWN for NSRs, since several studies have been conducted using an SFN for NSRs [19,20,21,22,27]. The physical infrastructure based on the SFN and SWN is being built using the nodes 100, 200, and 300, which are currently available.

Similarly to the previous work, the values for the NSR parameters were chosen at random from a predefined range of possibilities. The proposed technique was tested on two networks that operate under a variety of diverse situations. The number of NSRs was modified from 10 to 30, and the findings acquired after the algorithm was executed were recorded in a spreadsheet. Figure 6a depicts the resource efficiency of distinct NSRs that are provided under two different network configurations. It can be seen in the figure that the efficiency of network slicing is significantly improved when the network is organized according to the SWN topology. When the number of NSRs in a network is increased, the resource efficiency of both networks suffers a decline. The growth in the number of nodes in the infrastructure, on the other hand, enhances the efficiency in both circumstances. The resource efficiency is reported as 0.44 when using an SFN for serving 30 NSRs under a 100-node availability; however, the resource efficiency is recorded as 0.65 when using an SWN for serving 30 NSRs under a 100-node availability. As a result, an SWN-based network topology allows for better network slicing even during periods of high demand while using fewer resources. The SWN, on the other hand, performs much better as the amount of resources available is increased.

The performance of the algorithm in conjunction with the various network infrastructures was further tested by the acceptance ratio of the method. As shown in Figure 6b, the acceptance ratios of the SWN and SFN changed from 5 to 35 under different numbers of network nodes for provisioning (100, 200, and 300). It can be seen in the figure that an acceptance ratio of more than 0.9 was achieved in all of the instances. It does not matter what kind of network topology is used; as the demand increases, the acceptance ratio decreases. However, when comparing the provisioning via the SWN to the SFN structure under various physical nodes, the provisioning through the SWN delivers a much superior acceptance ratio. In the face of a peak demand of 35 NSRs with the bare minimum of resources, the SWN still manages to deliver a 0.68 acceptance ratio, whereas the SFN manages to produce just 0.62. Additionally, when the resources are expanded, the SWN yields an acceptance ratio of 0.77 for the peak demand. The SFN, on the other hand, achieves a 0.74 acceptance ratio with the same resources. A further validation of the proposed algorithm’s consistency was accomplished by running it through a series of 10 trials under the same operating circumstances and NSR values. For each number of NSRs, the minimum and maximum units of CPU serviced, as well as the bandwidth consumed, were logged during the 10 trials.

During execution, the method selects random values for all of the NSRs parameters, including the number of nodes, the CPU capacity, the bandwidth, the security level, and the lifespan, that are within their respective ranges of possibilities. The execution is carried out individually in the SFN and SWN network settings, with three node levels (100, 200, and 300) in each network configuration. Table 4 displays the outcomes that are achieved. From the table, it is evident that more CPU units are serviced and that more bandwidth is consumed when the network is organized according to the SWN topology. For 30 NSRs with 300 accessible nodes, an SWN-based network records the maximum values of CPU serviced and bandwidth consumed as 17,554 and 16,676, respectively, for the maximum values of CPU served and bandwidth utilized. The maximum values of CPU served and bandwidth consumed for the same circumstance are recorded by the SFN as 13,275 and 12,611, respectively.

The analysis was furthered by running the algorithm under three different numbers of NSRs with fixed lifetimes of the requests, with $LT_{min}$ = 10 and $LT_{max}$ = 40 as the starting and ending values. As demonstrated in Table 5, the resulting values of the CPU served and bandwidth consumed were recorded for NSRs, which ranged between 10 and 30 as shown in the table. It is obvious from the table that both forms of the physical structure record the maximum values of the CPU served and bandwidth consumed for LTmin requests, and vice versa for LTmax requests, and that this is true for both types of physical structure. The SWN structure, on the other hand, is able to share more resources than the SFN structure, as seen by the highest recorded value of 16,644 CPU served and 17,625 bandwidth consumed for 30 NSRs on the provided structure, which has 300 nodes in total. The SFN, on the other hand, could only serve 9322 CPUs while consuming just a 983 bandwidth for the same number of physical nodes. Furthermore, it is shown that the SWN may deliver superior services when compared to the SFN structure, and that service providers can earn a higher profit while simultaneously enhancing the utility.

## 6. Conclusions

The solution approach for effective network slicing in a 5G mobile network environment was developed in this study. The physical infrastructure, NSRs, and deployment plan were all modelled. To achieve a slicing-friendly infrastructure, physical infrastructure was represented as a small-world network. Physical infrastructure and NSR factors were found for efficient resource distribution. The deployment technique made use of PROMETHEE-II for node allocation and SLE for node-to-node connection creation. For the network slicing performance assessment, the resource efficiency and acceptance ratio were taken into account. The suggested algorithm’s performance was investigated under various network operating circumstances and varying NSR parameter values. The suggested algorithm’s competence was shown by comparing the result to other algorithms in the literature. Furthermore, to demonstrate the superiority of the small-world network (SWN), the suggested method was tested in a scale-free network (SFN) and a comparative analysis was performed. The findings reveal that the suggested algorithm operates better under various operating situations, and the small-world network provided a much better slicing provision. It is possible to develop this work to include the likelihood of the nodes’ placement, as the user cannot stand motionless. Machine and deep-learning-based node provisioning may also be advised for quick deployment. 

## Figures and Tables

**Figure 1 sensors-23-01556-f001:**
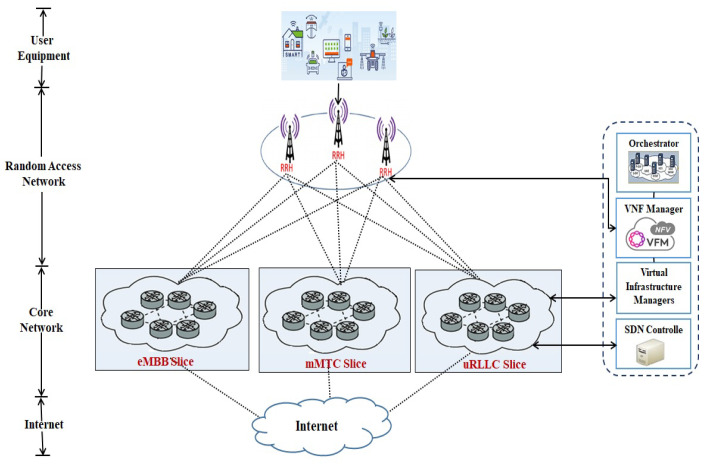
Architecture of 5G mobile network with slicing.

**Figure 2 sensors-23-01556-f002:**
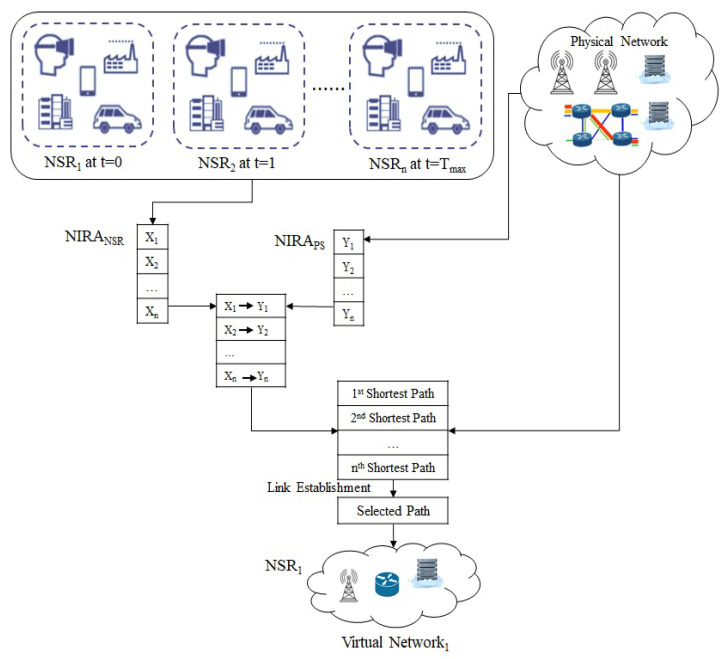
NSR deployment strategy with NIRA and SPA.

**Figure 3 sensors-23-01556-f003:**
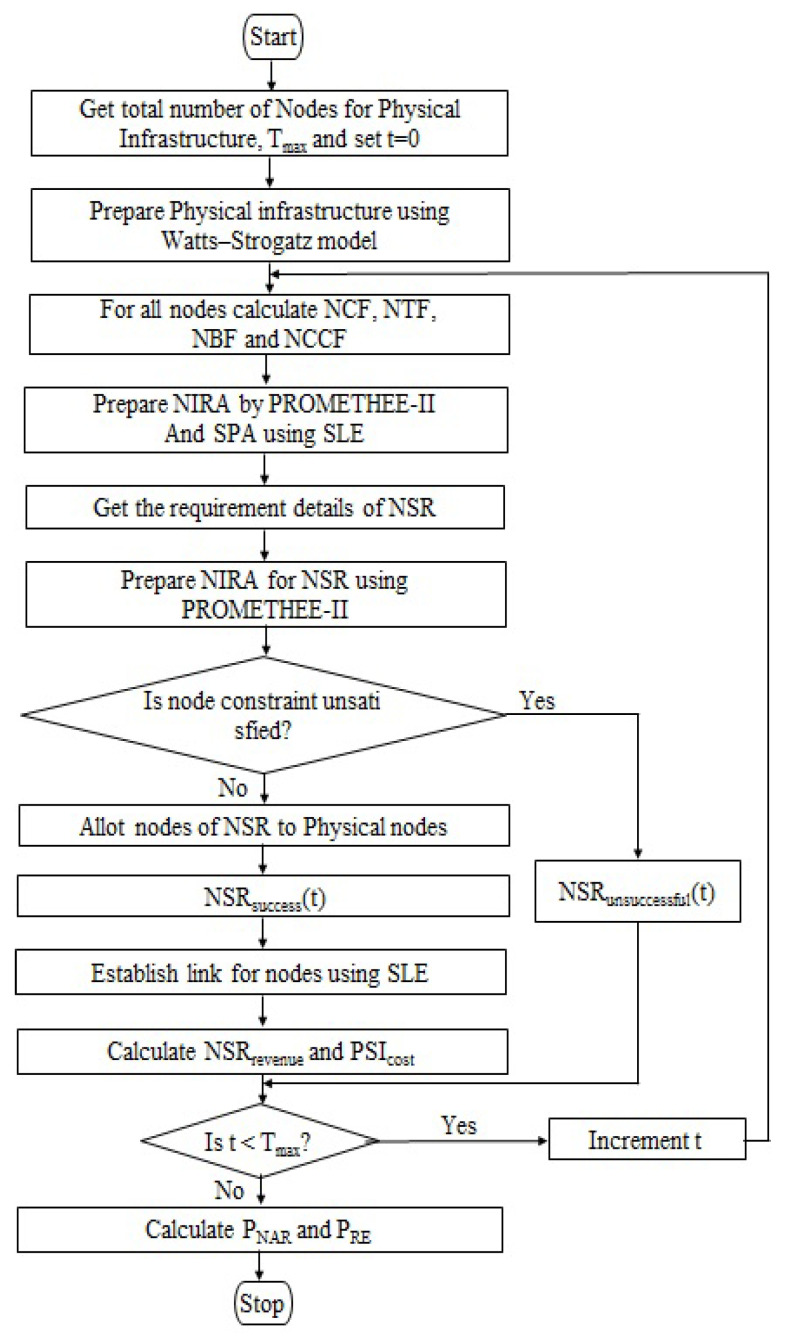
Depicts the sequence of events that occur throughout the whole solution method.

**Figure 4 sensors-23-01556-f004:**
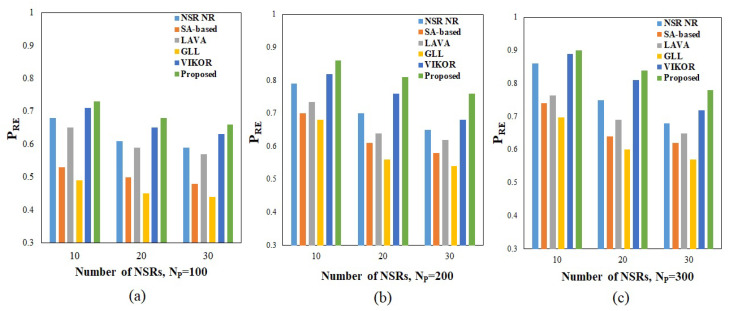
Resource efficiency under (**a**) 100, (**b**) 200, and (**c**) 300 nodes in physical infrastructure. Proposed method performance is compared with existing methods such as NSR-NR [19], SA [20], LAVA [21], GLL [22], VIKOR [27].

**Figure 5 sensors-23-01556-f005:**
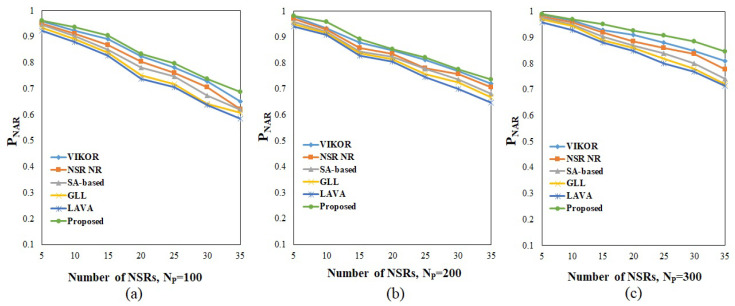
Acceptance Ratio under (**a**) 100, (**b**) 200, and (**c**) 300 nodes in physical infrastructure. Proposed method performance is compared with existing methods such as NSR-NR [19], SA [20], LAVA [21], GLL [22], VIKOR [27].

**Figure 6 sensors-23-01556-f006:**
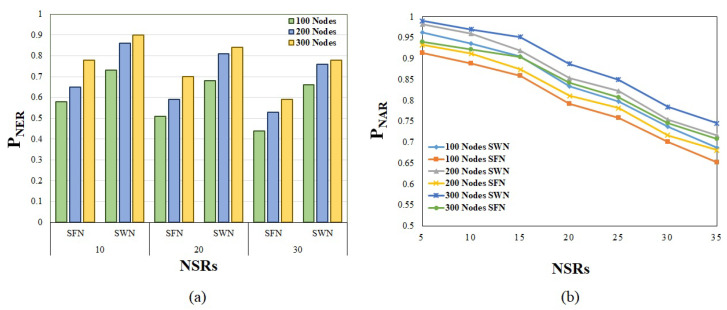
(**a**) Resource efficiency vs. number of NSRs node under number of nodes in physical infrastructure (100, 200, 300). (**b**) Acceptance ratio vs. number of NSRs node (10, 20, 30) under number of nodes in physical infrastructure (100, 200, 300).

**Table 1 sensors-23-01556-t001:** Summary of literature.

Ref. No.	Objectives	Node Provisioning for NSR Nodes	Link Provisioning for NSR Nodes	Physical Network	Resource	Security Issues
[17]	Slice acceptance ratio and revenue-to-cost ratio	Deterministic and random rounding	Deterministic and random rounding	Scale-free	Bandwidth and CPU capacity	Not considered
[19]	Acceptance ratio and resource efficiency	Acceptance ratio and resource efficiency	k shortest path Floyd algorithm	Scale-free	Bandwidth and CPU capacity	Not considered
[20]	Acceptance ratio and resource efficiency	Simulated annealing	Simulated annealing	Scale-free	Bandwidth and CPU capacity	Not considered
[21]	Acceptance ratio and resource efficiency	LAVA approach	LAVA approach	Scale-free	Bandwidth and CPU capacity	Not considered
[22]	Acceptance ratio and resource efficiency	GLL approach	GLL approach	Scale-free	Bandwidth and CPU capacity	Not considered
[23]	Slice acceptance ratio and revenue-to-cost ratio	Greedy node mapping	Dijkstra’s algorithm	Scale-free	Bandwidth and CPU capacity	Considered
[24]	Resource utilization and outage probability and resource efficiency	Markov decision process	Markov decision process	Scale-free NOMA	Power and subcarrier	Considered
[25]	Spectral efficiency and reliability	JSPA	JSPA	Scale-free OFDMA	Power and subcarrier	Not considered
[26]	Spectral efficiency and reliability	APSO	APSO	Scale-free OFDMA	Power and subcarrier	Not considered
[27]	Slice acceptance ratio and revenue-to-cost ratio	Node ranking using VIKOR	k shortest path algorithm	Scale-free	Bandwidth and CPU capacity	Considered
[28]	Slice classification accuracy	DBN and NN, GS-DHOA for weight function adjustments	DBN and NN, GS-DHOA for weight function adjustments	-	Performance dataset consists of network features	Considered
[29]	Latency and training loss	GNN model with DT	GNN model with DT	-	Performance dataset consists of network features	Considered
Proposed	Acceptance ratio and resource efficiency	Node ranking using PROMETHEE II	SPA formation through Dijkstra’s algorithm	Small-world network and scale-free network	Bandwidth and CPU capacity	Considered

**Table 2 sensors-23-01556-t002:** Evaluation table.

	$f_{1}\left(f_{\mathrm{NCF}}\right)$	$f_{2}\left(f_{\mathrm{NTF}}\right)$	$f_{3}\left(f_{\mathrm{NBF}}\right)$	$f_{4}\left(f_{\mathrm{NCCF}}\right)$
$n_{1}$	$f_{1}\left(n_{1}\right)$	$f_{2}\left(n_{1}\right)$	$f_{3}\left(n_{1}\right)$	$f_{4}\left(n_{1}\right)$
$n_{2}$	$f_{1}\left(n_{2}\right)$	$f_{2}\left(n_{2}\right)$	$f_{3}\left(n_{2}\right)$	$f_{4}\left(n_{2}\right)$
.	.	.	.	.
.	.	.	.	.
$n_{i}$	$f_{1}\left(n_{i}\right)$	$f_{2}\left(n_{i}\right)$	$f_{3}\left(n_{i}\right)$	$f_{4}\left(n_{i}\right)$

**Table 3 sensors-23-01556-t003:** Test case parameters.

Definitions	Descriptions	Range
$N_{P}$	The distribution of available security level of a node in real number	(0–1)
$CPU_{total}$	The distribution of CPU for each node in unit	U[30, 60]
$BW_{available}$	The distribution of bandwidth of each links in unit	U[30, 60]
L	The distribution of length of the links	U[1, 5]
$T_{NSR}$	The total number of NSRs arrived in the time frame	U[5, 35]
$N_{NSR}$	The distribution of nodes for each NSR	U[10, 30]
$CPU_{NSR}$	The distribution of CPU requirement	U[5, 25]
$BW_{NSR}$	The distribution of bandwidth requirement	U[5, 25]
$SPL_{NSR}$	The distribution of required security level of a node in real number	(0–0.5)
$LT_{NSR}$	The time duration of each NSR	T[10, 40]

**Table 4 sensors-23-01556-t004:** Performance comparison with $LT_{NSR}$ [$LT_{min}$ = 10, $LT_{max}$ = 40].

Model		100 Nodes				200 Nodes				300 Nodes			
		BW Utilized		CPU Served		BW Utilized		CPU Served		BW Utilized		CPU Served	
	**NSRs**	**Min**	**Max**	**Min**	**Max**	**Min**	**Max**	**Min**	**Max**	**Min**	**Max**	**Min**	**Max**
SFN	10	2328	4351	2212	4133	2605	4875	2501	4680	3123	5859	2967	5566
	20	4086	7653	3882	7270	4729	8852	4540	8498	5601	10508	5321	9983
	30	7927	9908	7531	9413	9544	11,925	9162	11,448	10,626	13,275	10,095	12,611
SWN	10	2921	5476	2775	5202	3440	6450	3302	6192	3605	6754	3425	6416
	20	5448	10,200	5176	9690	6484	12,156	6225	11,670	6720	12,600	6384	11,970
	30	11,880	14,852	11,286	14,109	13,680	17,102	13,133	16,418	14,049	17,554	13,347	16,676

**Table 5 sensors-23-01556-t005:** Performance comparison with fixed $LT_{NSR}$.

Model		100 Nodes				200 Nodes				300 Nodes			
		BW Utilized		CPU Served		BW Utilized		CPU Served		BW Utilized		CPU Served	
	**NSRs**	${\mathrm{LT}}_{\mathrm{Min}}$	${\mathrm{LT}}_{\mathrm{Max}}$	${\mathrm{LT}}_{\mathrm{Min}}$	${\mathrm{LT}}_{\mathrm{Max}}$	${\mathrm{LT}}_{\mathrm{Min}}$	${\mathrm{LT}}_{\mathrm{Max}}$	${\mathrm{LT}}_{\mathrm{Min}}$	${\mathrm{LT}}_{\mathrm{Max}}$	${\mathrm{LT}}_{\mathrm{Min}}$	${\mathrm{LT}}_{\mathrm{Max}}$	${\mathrm{LT}}_{\mathrm{Min}}$	${\mathrm{LT}}_{\mathrm{Max}}$
SFN	10	2406	914	2286	868	2811	1045	2699	1003	3321	1957	3155	1859
	20	6243	2431	5931	2309	7368	2692	7073	2584	7862	3164	7468	3006
	30	8683	5292	8248	5027	9321	7018	8948	6737	9813	7738	9322	7351
SWN	10	4812	1827	4571	1736	5622	2089	5397	2005	6642	3914	6310	3718
	20	10,486	8861	9862	8618	11,736	8384	11,147	8169	11,723	8328	10,937	8012
	30	13,365	10,584	12,497	10,055	15,641	11,035	14,895	10,874	17,625	12,476	16,644	10,702

## Data Availability

Not applicable.

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
