# Peer review of "Optimal Resource Allocation for 5G Network Slice Requests Based on Combined PROMETHEE-II and SLE Strategy"

_sensors, 2023, doi:10.3390/s23031556_

Round 1

Reviewer 1 Report

In this paper, a novel slicing technique is proposed to be employed for node allocation and connection establishment. Physical infrastructure and NSR factors were found for efficient resource distribution, and the deployment technique made use of PROMETHEE-II for node allocation and SLE for node-to-node connection creation.  

The introduction is well-written, a good overview of the related works is given, and the references are adequate. The contributions are stated in an appropriate way, and the algorithms are correctly described. Test case parameters are reasonably chosen. The numerical results obtained by simulation represent the significance of this approach.

It should be explained in more detail how the number of nodes (100,200 or 300) is determined, and why the values of SFN/SWN are set to (10, 20, and 30). It should be helpful if you can give a reference that indicates that these values are the typical ones.

Technical preparation of the paper has to be improved:

- if a sentence is finished with an equation, a dot has to be placed after the equation (e.g. in eqs. (1), (),...

- if the word 'where' is placed after the equation, usually a comma has to be placed after the equation (e.g. in eqs. (2), (19), (22-25),... Also, there is no need to place a comma after the word 'where';

- the first letters in some words in the titles of the journals in refs [3], [17], and [18] should be capitalized;

- the first letters in some words in the titles of the conferences in refs [8], [11] (please, compare it with [10]), and [21] should be capitalized;

- reference [24] is incomplete (vol, no, pp...).  

Author Response

We appreciate your insightful review remarks, which will help us improve our current work. We have responded to the comments to the best of our ability. We hope your expectations are met by the answers.

Reviewer 1

Comment 1: In this paper, a novel slicing technique is proposed to be employed for node allocation and connection establishment. Physical infrastructure and NSR factors were found for efficient resource distribution, and the deployment technique made use of PROMETHEE-II for node allocation and SLE for node-to-node connection creation.

The introduction is well-written, a good overview of the related works is given, and the references are adequate. The contributions are stated in an appropriate way, and the algorithms are correctly described. Test case parameters are reasonably chosen. The numerical results obtained by simulation represent the significance of this approach.

Response:

We thank you for taking the effort to review our work and provide the valuable feedback.

Comment 2: It should be explained in more detail how the number of nodes (100,200 or 300) is determined, and why the values of SFN/SWN are set to (10, 20, and 30). It should be helpful if you can give a reference that indicates that these values are the typical ones.

Response:

The NSR values that are being considered are 10, 20, and 30. Each NSR takes into consideration a total of 20 nodes. The physical infrastructure node counts are assumed to be 100, 200, and 300. The physical infrastructure and NSR values are based on the following reference.

References:

  1. Guan, W.; Wen, X.; Wang, L.; Lu, Z.; Shen, Y. A service-oriented deployment policy of end-to-end network slicing based on 518 complex network theory. IEEE access 2018, 6, 19691–19701.
  2. Mijumbi, R.; Serrat, J.; Gorricho, J. L.; Bouten, N.; De Turck, F.; Davy, S. Design and evaluation of algorithms for mapping 521 and scheduling of virtual network functions. In Proceedings of the Proceedings of the 2015 1st IEEE Conference on Network 522 Softwarization (NetSoft). IEEE, 2015, pp. 1–9.
  3. Li, X.; Guo, C.; Gupta, L.; Jain, R. Efficient and secure 5G core network slice provisioning based on VIKOR approach. IEEE Access 532 2019, 7, 150517–150529.

Also, we mentioned the citation in the paper's simulation and result section, which is section 5.1 (Page. No: 13, Ref. no: 19, 22, 27).

Comment 3: Technical preparation of the paper has to be improved:

- if a sentence is finished with an equation, a dot has to be placed after the equation (e.g. in eqs. (1), (),...

- if the word 'where' is placed after the equation, usually a comma has to be placed after the equation (e.g. in eqs. (2), (19), (22-25),... Also, there is no need to place a comma after the word 'where';

- the first letters in some words in the titles of the journals in refs [3], [17], and [18] should be capitalized;

- the first letters in some words in the titles of the conferences in refs [8], [11] (please, compare it with [10]), and [21] should be capitalized;

- reference [24] is incomplete (vol, no, pp...).

Response:

We have changed the respective sentences according to the recommendations of the reviewer.

Reviewer 2 Report

In this paper, the authors proposed a new approach to resource allocation for 5G network slice provisioning. 

Resource allocation for network slicing has been extensively studied with numerous methods proposed. Although the authors provided a review of some related works, the paper lacks a clear description of what limitation of the existing methods the proposed method can overcome; that is, what novelty the proposed method has and what advantage the proposed method may achieve over existing methods. Without sufficient descriptions of novelty, this research seems to be just another piece of work on a subject that has been well-studied. 

The authors highlighted the small world network model for the physical network infrastructure as a contribution of this paper. However, resource allocation for network slicing should not be limited to any network model of the underlying infrastructure (in other words, a desirable method for network slice provisioning should work for any physical infrastructure with an arbitrary network topology). The reviewer does not see the point of "developing a slicing-friendly physical design" as proposed n this research.  

The NSR model given in the paper is problematic. The paper claims that an NSR request is modeled as an undirected graph but the G_{NSR} is not a graph model -- a graph model G(V, E) has two sets: the set of vertices V and the set of edges E. The edge set is missing in G_{NSR}. 

In the proposed method, virtual node placement and routing are considered as two separate consecutive steps. However, the current trend of network-cloud/edge convergence makes it critical to jointly consider the computation and communication aspects in network slicing provisioning. The authors are suggested to refer to the article "Convergence of networking and cloud/edge computing: status, challenges, and opportunities" IEEE Network 34(6) and discuss the applicability of the proposed method in a converged network-cloud/edge environment. 

Author Response

We appreciate your insightful review remarks, which will help us improve our current work. We have responded to the comments to the best of our ability. We hope your expectations are met by the answers.

Reviewer 2

Comment 1: In this paper, the authors proposed a new approach to resource allocation for 5G network slice provisioning.

Resource allocation for network slicing has been extensively studied with numerous methods proposed. Although the authors provided a review of some related works, the paper lacks a clear description of what limitation of the existing methods the proposed method can overcome; that is, what novelty the proposed method has and what advantage the proposed method may achieve over existing methods. Without sufficient descriptions of novelty, this research seems to be just another piece of work on a subject that has been well-studied.

Response:

The PROMETHEE-II algorithm for allocating nodes and the SLE method for allocating links are being developed. The most important breakthrough in this work is the utilization of the small world network for the construction of physical infrastructure as well as NSR. The resource usage is improved with the aid of small world networks. The use of bandwidth and CPU resources in both scale-free networks and small world networks are compared. Utilizing a small world network results in improvements to bandwidth as well as CPU resources. It improves the performance of resource efficiency as well as the acceptance ratio.

Some content relevant to this has been included in Chapter 2. (Page no. 5)

Comment 2: The authors highlighted the small world network model for the physical network infrastructure as a contribution of this paper. However, resource allocation for network slicing should not be limited to any network model of the underlying infrastructure (in other words, a desirable method for network slice provisioning should work for any physical infrastructure with an arbitrary network topology). The reviewer does not see the point of "developing a slicing-friendly physical design" as proposed n this research.

Response:

The physical infrastructure has the scale-free network property, which says that when a communication network grows, an upcoming node has the tendency to connect itself to the nodes that already have a high degree, and the distribution of node degrees will follow a power law form. However, our objective is to examine the differences between the various infrastructure models to determine whether or not a small world network has benefits in 5G services. Small world networks have a trade-off in cost efficiency between high local clustering and short path lengths.

Compared to the scale-free network model, the small world model made more sense for our work. The small world network model enhances the efficiency of resource consumption. If resource utilization improved, user access rate increases automatically. Our small world-based PROMETHEE-II and SLE technique, which we presented, increases the performance of the model in terms of resource utilization when compared to the model that is previously used for resource allocation. It not only makes the infrastructure for the allocation of resources more convenient but also helps to improve the performance of the resources.

Comment 3: The NSR model given in the paper is problematic. The paper claims that an NSR request is modeled as an undirected graph but the G_{NSR} is not a graph model -- a graph model G(V, E) has two sets: the set of vertices V and the set of edges E. The edge set is missing in G_{NSR}.

Response:

Each undirected graph is made up of nodes and edges. Each node has a capacity, security power, and life time. Each edge consists of bandwidth. To avoid confusion, these specifications are described in more detail in Chapter 3.2 - NSR model (Page no. 5).

Comment 4: In the proposed method, virtual node placement and routing are considered as two separate consecutive steps. However, the current trend of network-cloud/edge convergence makes it critical to jointly consider the computation and communication aspects in network slicing provisioning. The authors are suggested to refer to the article "Convergence of networking and cloud/edge computing: status, challenges, and opportunities" IEEE Network 34(6) and discuss the applicability of the proposed method in a converged network-cloud/edge environment.

Response:

We thank the reviewer for the literature. In the revised manuscript we now discussed this issue in the related works. We included the following reference based on your comments in Chapter 2 – Related works Section (Page no. 3).

Reference:

Duan, Q., Wang, S., & Ansari, N. (2020). Convergence of networking and cloud/edge computing: Status, challenges, and opportunities. IEEE Network, 34(6), 148-155.

Round 2

Reviewer 2 Report

The authors have made the revisions that address each item of the reviewer's comments. Although still has different options regarding the suitability of the small-world network as the network infrastructure, the reviewer believes that this paper offers some interesting insights that are worth reading, so recommends its publication.